# Interplay between Task Learning and Skill Discovery for Agile Locomotion

## Abstract

Agile locomotion of legged robots, characterized by high momentum and frequent contact changes, is a challenging task that demands precise motor control. Therefore, the training process for such skills often relies on additional techniques, such as reward engineering, expert demonstrations, and curriculum learning. However, these requirements hinder the generalizability of methods because we may lack sufficient prior knowledge or demonstration datasets for some tasks. In this work, we consider the problem of automated learning agile motions using its intrinsic motivation, which can greatly reduce the effort of a human engineer. Inspired by unsupervised skill discovery, our learning framework encourages the agent to explore various skills to maximize the given task reward. Finally, we train a parameter to balance the two distinct rewards through a bi-level optimization process. We demonstrate that our method can train quadrupeds to perform highly agile motions, ranging from crawling, jumping, and leaping to complex maneuvers such as jumping off a perpendicular wall.

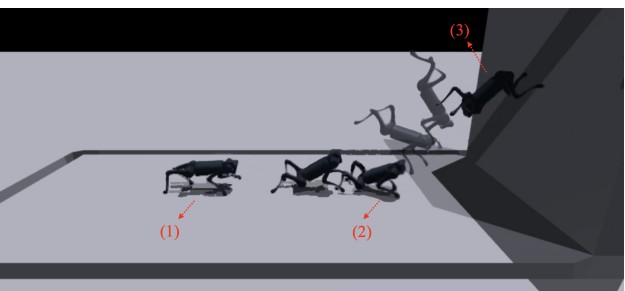

Figure 1: A figure showing highly agile behavior trained using our method. The quadruped is (1) running toward the wall, (2) jumping off the ground and performing a front flip clockwise, and (3) using its hind legs to kick the perpendicular wall, rotating counterclockwise, and landing on the ground.

## 1 Introduction

Agile motor skills are challenging for both humans and robots to learn because they require complex planning of full-body movements and precise motor control. For example, mastering advanced gymnastics skills involves carefully coordinating the teaching and practice phases. Some unintuitive motor skills, such as the Fosbury flop in high jump or the Eurostep in basketball, took athletes decades to discover. Similarly, developing controllers for agile skills remains one of the most difficult tasks in robotics, which makes an algorithm easily get stuck in local minima.

In recent years, legged robot locomotion, when combined with deep reinforcement learning (RL), has reached a high level of agility (Tan et al., 2018; Lee et al., 2020b; Hwangbo et al., 2019; Xie et al., 2021; Song et al., 2020; Haarnoja et al., 2018; Smith et al., 2023; Luo et al., 2024). However, those algorithms often require additional techniques to learn challenging skills, such as reward engineering based on domain expertise (Zhuang et al., 2023; Cheng et al., 2023; Yang et al., 2023b), demonstration datasets (Bogdanovic et al., 2022; Kilinc & Montana, 2022; Li et al., 2023a; He

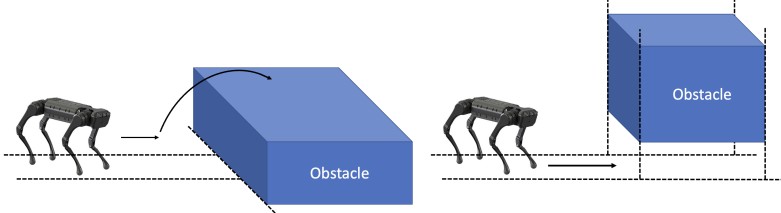

Figure 2: A robot must explore diverse strategies to overcome the obstacle from a simple task description, such as jumping over (**Left**) or crawl under (**Right**).

et al., 2024), or carefully designed curriculum learning (Kumar et al., 2021). This paper's goal is to develop an automated learning algorithm that reduces manual engineering effort, which can learn highly agile skills such as the *wall-jump* shown in Figure 1. However, developing an automated learning framework for agile locomotion is not straightforward because the high momentum and contact changes make an optimization ill-conditioned with multiple local minima.

In this work, we aim to design a learning algorithm to achieve the given difficult task by exploring a diverse set of possible approaches. Consider a locomotion task with two types of obstacles, as shown in Figure 2: the robot must jump over the obstacle in the left figure and crawl under the box in the right figure. We want the robot to overcome both tasks based on a simple task description, such as "moving forward." However, the robot would struggle to solve these tasks because this description does not provide enough incentive to explore different base heights. Therefore, our algorithm must intrinsically motivate the robot to examine various gaits, especially when learning with the given task reward becomes saturated.

In detail, our approach combines two objectives: solving the given task and finding diverse solutions. Solving the task is represented by maximizing the task reward. The task reward should be kept simple, such as following forward velocity commands to move toward task completion. On the other hand, exploring diverse behaviors is achieved by maximizing a diversity reward, which is derived from skill discovery methods. This encourages the agent to try various approaches to find the desired height, orientation, velocity, or angular velocity needed to solve the task. However, balancing two distinct objectives is not straightfoward and one may overpower the other. If the task reward dominates, agents may not sufficiently explore diverse behaviors. Conversely, if the diversity reward dominates, agents may spend too much time exploring, failing to solve the task. This is analogous to the exploration-exploitation trade-off in RL (Sutton, 2018). To address this problem, we introduce a learnable parameter $\lambda$ to balance the two objectives. We train $\lambda$ to automatically adjust the weight of the diversity reward to maximize the task reward. Details of training $\lambda$ will be covered in Section 3.2. This approach enables the agent to effectively balance exploration and exploitation.

In summary, our approach aims to adopt skill discovery methods to enhance the task-specific reward by incorporating human priors. The primary contributions of this work are as follows: (1) We propose a novel framework that combines RL and unsupervised skill discovery algorithms to automatically learn agile locomotion skills. (2) We provide a thorough derivation of bi-level optimization framework for training the balancing parameter $\lambda$. We also demonstrate that our approach of adapting $\lambda$ robustly finds the optimal value for a given task. (3) We evaluate our method on three challenging locomotion tasks: jumping, leaping, and crawling. In these environments, we compare our approach against exploration-based methods for utilizing human priors, showing that our method outperforms the baselines. (4) We demonstrate that our method can train unprecedented levels of agile behavior, such as accomplishing a wall-jump.

## 2 RELATED WORK

**Unsupervised Skill Discovery.** To establish an association between the skill $z$ and the resulting policy $\pi(a|s, z)$, DIAYN (Rudin et al., 2022a) proposes maximizing the mutual information between skills and states, $I(z; s)$. However, a limitation of DIAYN is that its objective can be fully optimized

with only minor differences between states, as long as the discriminator can distinguish between the skills, even if these differences are minimal.

To address this issue, LSD (Park et al., 2022) suggests an alternative objective that provides more incentive to increase state differences. However, LSD measures state differences using Euclidean distance, which leads to a focus on "easy change" within existing state dimensions. For example, in manipulation tasks, changing the robot arm's end-effector position is considered an easy-to-change state, whereas altering the target object's position is more challenging. To tackle this challenge, CSD (Park et al., 2023a) introduced a different distance metric called "controllability-aware distance". This metric assigns higher values to state transitions that are less likely to occur, thereby encouraging the learning process to focus more on state changes that are rare.

It is worth noting that DIAYN, LSD, and CSD primarily address low-dimensional state spaces. METRA (Park et al., 2023b), on the other hand, tackles the skill discovery problem in high-dimensional image inputs. METRA also incorporates the Wasserstein Dependency Measure, $I_{WDM}$ (Ozair et al., 2019), between skill $z$ and states, encouraging the agent to visit maximally different states for different skills, based on the given distance metric. We utilized METRA as our base skill discovery algorithm.

**Learning Agile Locomotion.** Recently, learning-based methods have demonstrated highly agile locomotion capabilities such as high-speed running (Margolis et al., 2022; Fu et al., 2021), jumping (Li et al., 2023b; Yang et al., 2023a), and climbing (Rudin et al., 2022a; Lee et al., 2020a). Our work aims to cover not only jumping, running, and leaping, but also *wall-jumping*, which involves a parkour-style motion combining flipping and jumping using walls.

The work most related to ours is that of Zhuang et al. (2023), which used a manually designed reward that penalizes the overlap between the robot and imaginary obstacles. They trained agents to minimize these overlaps, resulting in the learning of agile behaviors. In contrast, we aim to train a similar set of tasks without the need for extensive reward designs. Instead, we allow an unsupervised reinforcement learning (RL) method to discover the skills required to solve these tasks.

## 3 METHOD

### 3.1 PROBLEM FORMULATION

We regard the problem of training a control module of a legged robot as a Markov Decision Process (MDP) defined as $\mathcal{M} \equiv \{\mathcal{S}, \mathcal{A}, \mathcal{R}, \mathcal{P}, \gamma\}$, where $\mathcal{S}$ is a state space, $\mathcal{A}$ is action space composed of joint torques of the robot, $\mathcal{R}$ is a reward function, $\mathcal{P}$ is a transition probability, and $\gamma$ is a discount factor. When given a specific MDP, RL offers a way of obtaining an optimal policy $\pi$ which maximizes the expected sum of the discounted reward $J = \mathbb{E}_\pi \left[ \sum_{t=0}^{\infty} \gamma^t r_t \right]$. $\pi$ can be parameterized with neural network $\theta$, so here we denote policy as $\pi_\theta$.

### 3.2 OUR APPROACH

Overall, instead of a standard policy $\pi_\theta(a|s)$, we train a skill-conditioned policy $\pi_\theta(a|s, z)$, where $z$ is randomly sampled from a prior distribution, $z \sim p(z)$, for each episode and remains fixed throughout the episode. Our objective is to find $\theta$ that optimizes the expected sum of both the task reward $r^{\text{task}}$ and the diversity reward $r^{\text{div}}$.

$$\theta = \arg \max_{\theta} J^{\text{task+div}} = \arg \max_{\theta} \mathbb{E}_{\pi_\theta} \left[ \sum_{t=0}^{\infty} \gamma^t (r_t^{\text{task}} + \lambda r_t^{\text{div}}) \right]$$

A learnable parameter $\lambda$ determines the weight of $r^{\text{div}}$, and we refer to it as the balancing parameter. The task reward $r_t^{\text{task}}$ specifies the goal of the task. It can be defined for each task and should be kept simple, such as a velocity-following or forward-movement reward. Regardless of the value of $\lambda$, the policy $\pi$ is always conditioned on a particular $z$. Conditioning the policy on different values of $z$ results in different behaviors, so training a skill-conditioned policy with $\lambda = 0$ effectively means we are training a group of different policies, all of which converge into a single behavior. When $\lambda$

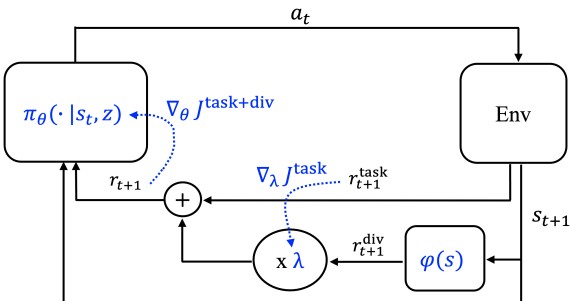

Figure 3: A figure of bi-level optimization for $\pi_\theta$ and $\lambda$. Task reward gives the gradient signal for training $\lambda$, and sum of two sources of rewards provides the gradient signal for optimizing $\pi_\theta$.

becomes large, the diversity reward dominates, and each policy learns a distinct skill, but none of them are capable of solving the task. Thus, determining the appropriate value of $\lambda$ is crucial. In the following section, we will explain how the balancing parameter $\lambda$ is trained and how $r^{\text{div}}$ is defined.

**Train Balancing Parameter**   As depicted in the Figure 3, we utilize a bi-level optimization framework to train both policy $\pi$ and a learnable balancing parameter $\lambda$, which is similar to LIRPG (Zheng et al., 2018). While $\theta$ is trained to maximize $J^{\text{task+div}}$, $\lambda$ is trained to maximize only $J^{\text{task}} = \mathbb{E}_{\pi_\theta}\left[ \sum_{t=0}^{\infty} \gamma^t r_t^{\text{task}} \right]$. It is worth noting that our ultimate goal is to solve the external task. So the intuitive meaning of training $\lambda$ solely depending on the task reward is that we determine the degree of diversity reward only to maximize the task performance. Ideally, when diversity reward helps solve the task, $\lambda$ will be increased, and if it rather deters training, $\lambda$ will be decreased.

More concretely,

$$\lambda = \arg\max_\lambda J^{\text{task}} \tag{1}$$

The problem here is we cannot directly compute the gradient of $J^{\text{task}}$ against $\lambda$, so we use the chain rule to compute the gradient of $\lambda$ with respect to $J^{\text{task}}$.

$$\nabla_\lambda J^{\text{task}} = \nabla_{\theta'} J^{\text{task}} \nabla_\lambda \theta' \tag{2}$$

Here, we can compute the first term $\nabla_{\theta'} J^{\text{task}}$ using policy gradient theorem (Sutton et al., 1999)

$$\nabla_{\theta'} J^{\text{task}} \approx A^{\text{task}} \nabla_{\theta'} \log \pi_{\theta'}(a|s, z) \tag{3}$$

where $A^{\text{task}}$ and $A^{\text{div}}$ refers to the advantage value computed with $r^{\text{task}}$ and $r^{\text{div}}$ respectively, and corresponding value functions $v_{\psi_1}^{\text{task}}$ and $v_{\psi_2}^{\text{div}}$. To compute the second term $\nabla_\lambda \theta'$, we first derive $\theta'$.

$$\theta' = \theta + \alpha \nabla_\theta J^{\text{task+div}}(\theta)$$
$$= \theta + \alpha A^{\text{task+div}} \nabla_\theta \log \pi_\theta(a|s, z) \tag{4}$$

Then we can plug in this result to compute $\nabla_\lambda \theta'$:

$$\nabla_\lambda \theta' = \nabla_\lambda (\theta + \alpha A^{\text{task+div}} \nabla_\theta \log \pi_\theta(a|s, z))$$
$$= \nabla_\lambda (\alpha A^{\text{task+div}} \nabla_\theta \log \pi_\theta(a|s, z))$$
$$= \nabla_\lambda (\alpha A^{\text{task}} + \alpha \lambda A^{\text{div}}) \nabla_\theta \log \pi_\theta(a|s, z)$$
$$= \alpha A^{\text{div}} \nabla_\theta \log \pi_\theta(a|s, z) \tag{5}$$

Finally, we can compute the value of $\nabla_\lambda J^{\text{task}}$ by pluggin in the Eq. (3) and Eq. (5):

$$\nabla_\lambda J^{\text{task}} \approx A^{\text{task}} \nabla_{\theta'} \log \pi_{\theta'}(a|s, z) * \alpha A^{\text{div}} \nabla_\theta \log \pi_\theta(a|s, z) \tag{6}$$

We can compute this term using sample-based approximation. The difference between our approach and Zheng et al. (2018) is that instead of training the intrinsic reward function itself, we fix the intrinsic reward as the diversity reward, and we only train the balancing parameter $\lambda$ to determine the degree of it.

**Diversity Reward** For the diversity reward $r^{\text{div}}$, we follow the formulation of METRA (Park et al., 2023b). They train skills to maximize Wasserstein Dependency Measure (Ozair et al., 2019) $I_{\text{WDM}} = I_W(S; Z)$. Maximization of the $I_{\text{WDM}}$ can be translated into following objective:

$$\sup_{\pi, \phi} \mathbb{E}_{P(\tau, z)} \left[ \sum_{t=0}^{T-1} (\phi(s_{t+1}) - \phi(s_t))^T z \right] \text{ s.t. } \|\phi(s) - \phi(s')\|_2 \leq 1, \forall (s, s') \in \mathcal{S}_{\text{adj}},$$

Here, $\phi : S \to Z$ is a learnable representation function that maps state into latent skill space. Optimization of this term can be achieved by simply using the off-the-shelf RL algorithm to maximize the reward $r^{\text{div}} = (\phi(s_{t+1}) - \phi(s_t))^T z$. To ensure that $\phi$ satisfies the constraint, we use dual gradient descent with a Lagrange multiplier $\kappa$ with a small margin $\epsilon > 0$. Please refer to Park et al. (2023b) for more details.

**Skill Selection** A typical unsupervised skill discovery method requires careful selection of the skill vector $z$ during the testing phase. However, we observed that as training progresses, an increasing proportion of the learned skills exhibit successful behaviors, a phenomenon we refer to as "positive collapse" (Section 4.3). Therefore, in this work, we simply select a random skill $z$ for reporting performance, rather than selectively choosing it or training a high-level controller.

**Implementation Details** We introduced two separate value networks, $v^{\text{task}}\psi_1$ and $v^{\text{div}}\psi_2$, due to the presence of two distinct reward sources: $r^{\text{task}}$ and $r^{\text{div}}$. Using a single value network to model the value of $r^{\text{task}} + \lambda r^{\text{div}}$ led to unstable training, as the scale of the rewards varied with changes in $\lambda$. Pseudo-code for our algorithm is provided here.

---

**Algorithm 1**

---

1: Initialize skill-conditioned policy $\pi_\theta(a|s, z)$, value functions $v_{\psi_1}^{\text{task}}$ and $v_{\psi_2}^{\text{div}}$, representation function $\phi(s)$, Lagrange multiplier $\kappa$, Balancing parameter $\lambda$, data buffer $\mathcal{D}$
2: **for** $i \leftarrow 1$ to # of epochs **do**
3:    **for** $j \leftarrow 1$ to # of episodes per epoch **do**
4:       Sample skill $z \sim \mathcal{N}(0, I)$
5:       **while** episode not terminates **do**
6:          Sample action $a \sim \pi(a|s, z)$
7:          Execute $a$ and receive $s'$ and $r^{\text{task}}$
8:          Compute $r^{\text{div}} = (\phi(s') - \phi(s))^T z$
9:          Add $\{s, a, r^{\text{task}}, r^{\text{div}}, s'\}$ to data buffer $\mathcal{D}$
10:       **end while**
11:    **end for**
12:    **for** $\{s, a, r^{\text{task}}, r^{\text{div}}, s'\}$ in $\mathcal{D}$ **do**
13:       Update $\phi(s)$ to maximize $\mathbb{E}_{(s,z,s') \sim \mathcal{D}} \left[ (\phi(s') - \phi(s))^T z + \kappa \cdot \min(\epsilon, 1 - \|\phi(s) - \phi(s')\|_2^2) \right]$
14:       Update $\kappa$ to minimize $\mathbb{E}_{(s,z,s') \sim \mathcal{D}} \left[ \kappa \cdot \min(\epsilon, 1 - \|\phi(s) - \phi(s')\|_2^2) \right]$
15:       Update $\theta$ using PPO with reward $r = r^{\text{task}} + \lambda * r^{\text{div}}$
16:       Update $\psi_1$ and $\psi_2$ using $r^{\text{task}}$ and $r^{\text{div}}$ respectively
17:       Update $\lambda$ using Eq. 6
18:    **end for**
19: **end for**

---

## 4 EXPERIMENTAL RESULTS

In this section, we evaluate the proposed framework by training policies on a set of agile locomotion tasks. First, we examine three robot parkour learning tasks from Zhuang et al. (2023), including climbing, crawling, and leaping, which require distinctive control strategies to overcome obstacles. On these tasks, we experiment with how skill discovery methods can aid in learning agile behaviors and evaluate our methods against baselines. Next, we investigate the effect of learning an adjustable $\lambda$, and compare performance against trials with fixed value of $\lambda$ value. We then show that all diverse skills are converged to a single optimum skill. Finally, we push our method to its limits in terms of agility to explore the most agile motions it can learn.

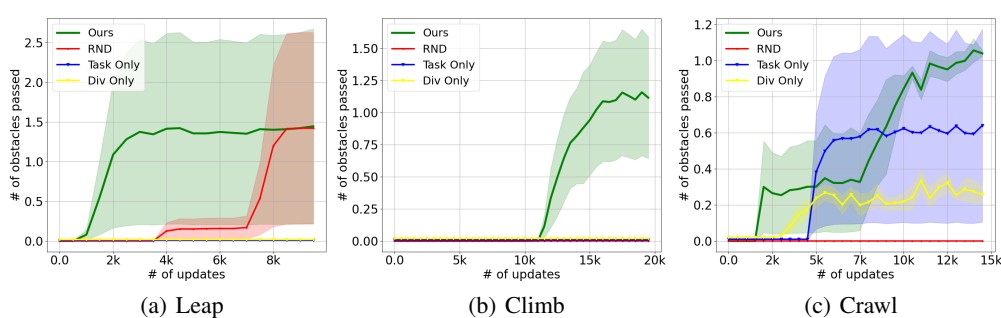

Figure 4: Training curve of our methods against baseline algorithms on three different tasks. Our method can solve all the tasks and exhibits better sample efficiency. Three different seeds were used.

**Simulation Setup**  We use Isaac Gym (Makoviychuk et al., 2021) as a simulation engine. Our codebase is developed based on the work of Rudin et al. (2022b). We use the Unitree A1 robot for all our experiments. The observation space is detailed in Appendix A.1. We use Proximal Policy Optimization (PPO) (Schulman et al., 2017) as our main RL algorithm. Our policies converge in 10k–20k iterations depending on the task, which takes 8–16 hours on an NVIDIA A40 GPU.

## 4.1 Learning Agile Locomotion Skills

We compared our method against the following baseline algorithms:

- *Task-only*: An RL baseline trained only with task specific rewards $r^{\text{task}}$.
- *Div-only*: An RL baseline trained using diversity reward $r^{\text{div}}$ only.
- *RND* (Burda et al., 2018): It combines $r^{\text{task}}$ with an exploration reward instead of a diversity reward.

We designed the same task reward across all baseline methods and tasks, with the primary goal of incentivizing agents to move forward. Details of the task rewards are provided in Appendix A.2. For both the diversity reward and exploration bonus in RND, we manually specify sub-dimensions of the state space, ensuring that the learning process focuses on diversity within the specified sub-dimensions. Specifically, we selected base heights for climbing and crawling tasks and forward velocity for leaping. Additionally, to expedite the learning of the *Div-only* agent, we provided the robot's base $x$ position as additional input to the skill discovery algorithm. This facilitated the exploration of diverse $x$ positions, ultimately helping the agent move forward.

**Our method enables the effective learning of agile motions.**  We present the training curves of our method and all baseline algorithms in Figure 4. We measured the number of obstacles passed in each task, where each task contains three consecutive obstacles of same configuration. Our method successfully learned the necessary motor skills for all tasks. Compared to the *Task-only* baseline, we observed that incorporating diversity rewards helps in learning agile locomotion skills. However, relying solely on diversity rewards (*Div-only*) fails to achieve meaningful skills, highlighting that a balanced interplay between task and diversity rewards is critical for success. Additionally, a comparison with *RND* shows that diversity-based approaches outperform exploration-based rewards. We believe this is because exploration-based methods focus on 'local' exploration, incentivizing agents to visit nearby unvisited states, making 'global' exploration challenging. In contrast, skill discovery methods inherently facilitate global exploration, as they encourage skills to explore distinct sets of states, allowing agents to transition to entirely new regions.

**Skill discovery enables high level exploration.**  We also provide qualitative evidence demonstrating how skill discovery methods enhance exploration. Figure 5 illustrates example behaviors of our method using two different skills for each task based on an actual model checkpoint from training. To observe the behaviors of different skills, we kept the model fixed and fed different skill vectors

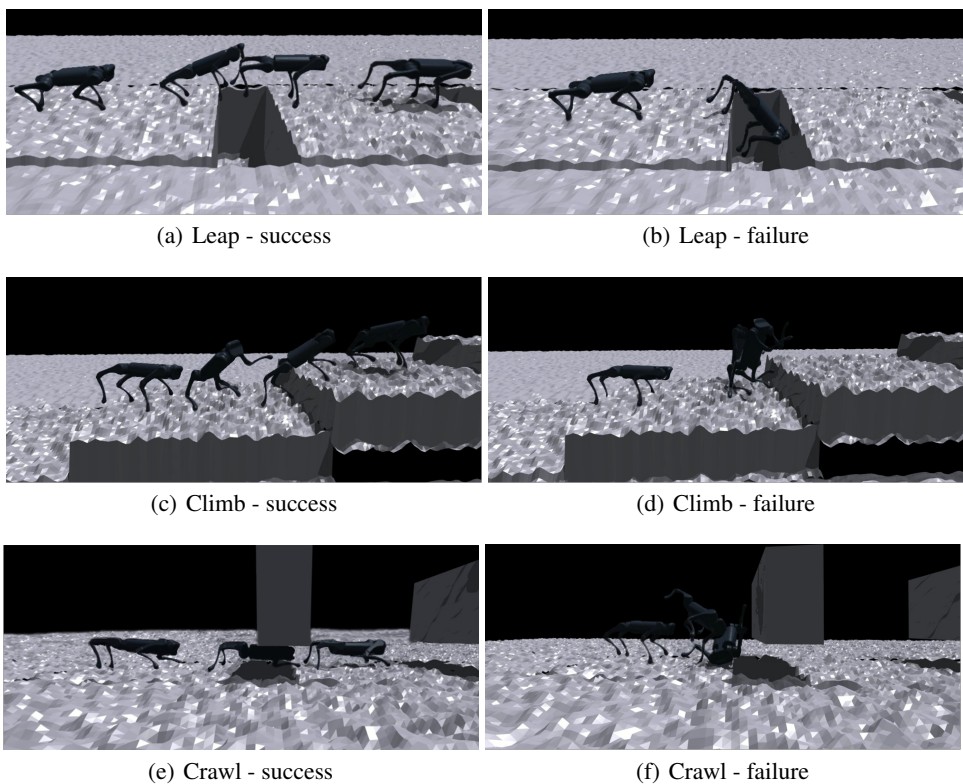

(a) Leap - success        (b) Leap - failure

(c) Climb - success        (d) Climb - failure

(e) Crawl - success        (f) Crawl - failure

Figure 5: Visualization of the diverse skills explored by the robot during training. For each task, we used the same model checkpoint but applied different skills to generate rollouts for both successful and failed cases.

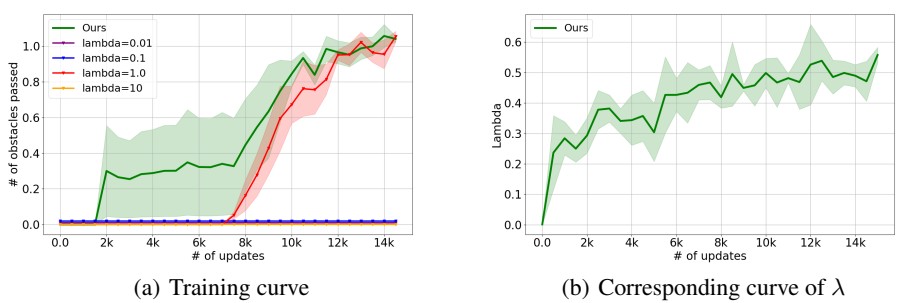

(a) Training curve        (b) Corresponding curve of $\lambda$

Figure 6: Our method outperforms all the baseline rewards with fixed value of lambda.

to the policy. As a result, both successful and unsuccessful episodes were generated from the same policy, using different skill vectors. In the crawling task, some skills successfully navigated past the obstacle, while others crashed and lost balance. Similarly, in the leaping task, certain skills allowed the agent to jump over the gap, whereas others failed and fell. The climbing task shows a similar variation. These examples confirm that the agent explores diverse behaviors; some of which solve the task while others do not. When a particular skill starts solving the task, the task reward increases, leading to successful task completion. In this sense, skill discovery functions as a high-level exploration module.

| Leap | | | Climb | | | Crawl | | |
|------|------|------|-------|------|------|-------|------|------|
| 1k | 2k | 3k | 12k | 15k | 20k | 2k | 7k | 15k |
| 29.9±5.2 | 99.1±0.9 | 99.4±0.8 | 49.4±7.2 | 71±9.3 | 68.7±11 | 22.3±3.1 | 31.7±3.8 | 40±2.8 |

Table 1: Ratio of successful skill vectors $z$ for each checkpoint (%)

## 4.2 LEARNING BALANCING PARAMETER $\lambda$

Selecting the appropriate value for $\lambda$ is crucial, as the scale of both the task reward and diversity reward is difficult to determine a priori. If either the task reward or the diversity reward dominates, the agent's learning process can be significantly hindered. In this section, we demonstrate how our algorithm effectively adjusts $\lambda$ during training. We compare our adaptive approach to fixed values of $\lambda$, using four different settings: $0.01, 0.1, 1, 10$. These experiments were conducted on the crawling tasks from the previous section, with each method trained using three different random seeds. We measured performance based on the number of obstacles passed.

**Our method outperforms fixed $\lambda$ values.** Figure 6(a) shows that our adaptive method outperforms all fixed-value experiments. Training with fixed $\lambda$ values of $0.01, 0.1$, and $10$ failed to pass a single obstacle, while both our method and the fixed $\lambda$ of $1.0$ successfully solved the task. However, our approach demonstrated superior sample efficiency compared to $\lambda = 1.0$. Figure 6(b) illustrates how the learned $\lambda$ values evolve during training. The value starts at 0 and gradually increases, suggesting that our algorithm learned that increasing $\lambda$ helps maximize task rewards over time. Additionally, Appendix Figure 8 shows the evolution of $\lambda$ for all three parkour tasks. The results indicate that some tasks require a gradual increase in $\lambda$, while others benefit from maintaining a steady value in the range of $[0.2, 0.4]$.

It is also important to note that our method does not correspond to a single fixed $\lambda$ value throughout training. In other words, there may not exist a single value of $\lambda$ that could yield an identical training curve. Our approach adjusts $\lambda$ dynamically, resulting in different values at different stages of training, which allows the agent to achieve an appropriate balance of diversity and task reward throughout the learning process.

## 4.3 CONVERGENCE OF DIFFERENT SKILLS INTO A NARROW SOLUTION SPACE

One potential challenge of incorporating a skill discovery module into the learning process is the difficulty of selecting the exact skill that solves the task after training, especially if only a small portion of the skill space is effective. However, we observed that as training progresses, a growing number of skill vectors $z \sim \mathcal{N}(0, I)$ become capable of solving the task. To demonstrate this, we selected model checkpoints at various stages of training and measured the success rate using 100 randomly sampled skills. This experiment was repeated ten times to determine the standard deviation.

The results are presented in Table 1. For the leap task, initially, only 30% of the skills were successful, but this number eventually approached nearly 100%. Similarly, for the climb and crawl tasks, the proportion of successful skills increased steadily. This suggests that once a viable solution is discovered, different skill vectors converge into similar behaviors with the solution, especially when the solution space is narrow for the given task. This contrasts with a typical skill discovery scenario where only a small subset of skills solves the task. Instead, in our case, the proportion of successful skills increased significantly over time.

This indicates that the resulting behavior of $\pi(a|s, z)$ can converge to a similar behavior despite using different skill vectors $z$. Intuitively, the training process involves an initial phase of exploration, followed by convergence to a solution. We observe that this phenomenon of later convergence is facilitated by task rewards: when a skill finds a successful solution, the corresponding trajectory receives higher rewards, which results in the increased probability of the actions taken. Because all skills share the same policy network, this learning propagates to other skill-conditioned behaviors, leading to what we term a "positive collapse" of skills. Initially diverse behaviors converge to a

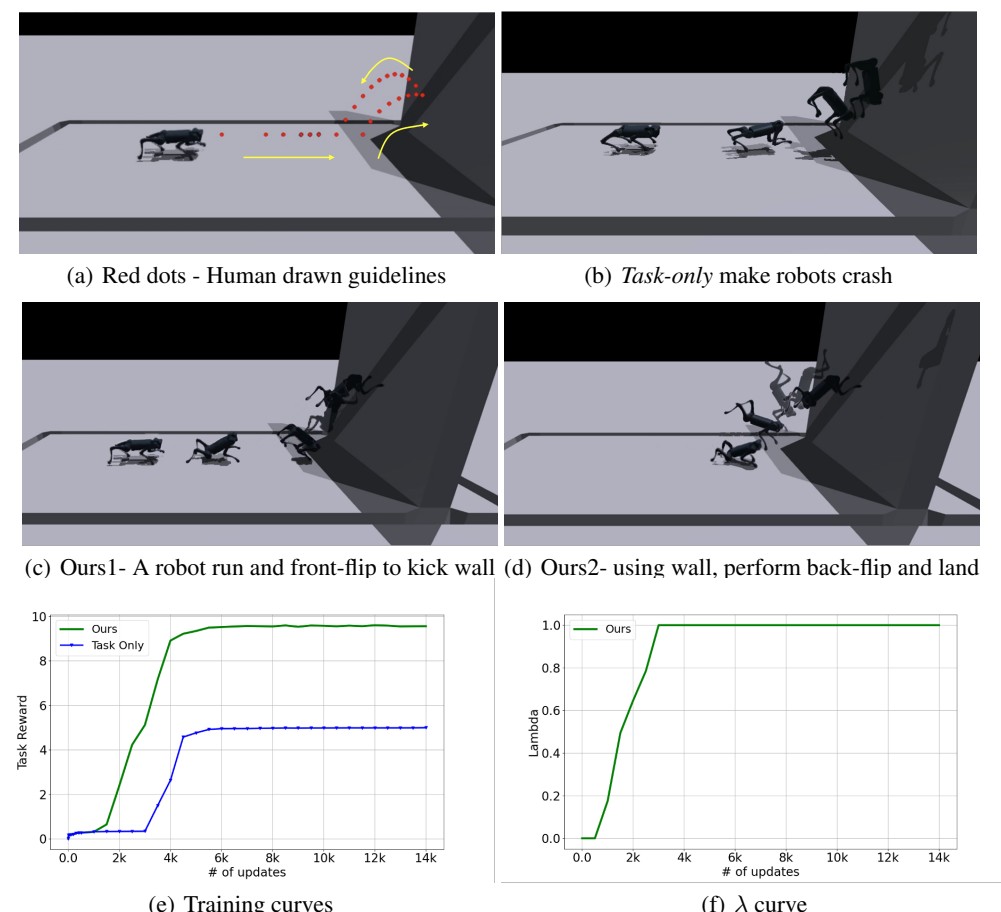

(a) Red dots - Human drawn guidelines

(b) *Task-only* make robots crash

(c) Ours1- A robot run and front-flip to kick wall

(d) Ours2- using wall, perform back-flip and land

(e) Training curves

(f) $\lambda$ curve

Figure 7: Our method enables robots solve wall-jump task.

common, successful strategy, which is beneficial as it maximizes task rewards and eliminates the need to manually select the right skill.

### 4.4 WALL-JUMP : LEARNING SUPER AGILE TASKS

Lastly, we pushed our method to its limits. We introduced a new task named *wall-jump*, which requires the robot to perform a sequence of highly agile motions, including running, jumping, flipping, and landing in a specific order. To make this feasible, we devised a guideline-based reward that is widely adopted in robotics(Tang et al., 2021; Gu et al., 2023). The reward encourages the agent to follow the guideline specified by a user. We used this reward as $r^{\text{task}}$. More details about the reward design can be found in Appendix C. The exact guideline used is shown in Figure 7(a). Note that the guideline only provides the target trajectory for the root position while not offering any information about orientation.

However, providing the guideline alone was not sufficient for the agent to successfully perform the wall-jump. Figure 7(b) shows the resulting behavior of the agent trained solely with $r^{\text{task}}$. The robot was able to follow the guideline up until it reached the perpendicular wall, but then crashed its back against the wall. The cumulative reward for this episode was about 5.0, as shown by the blue curve in Figure 7(e). We observe that the robot needs to acquire a specific orientation to kick off the wall and land safely.

Therefore, we provided the robot's base's `roll`, `pitch`, and `yaw` as input to the skill discovery algorithm, allowing our method to explore and learn diverse orientations of the robot when needed. Figures 7(c) and (d) show the resulting behavior. Our method was able to acquire the specific

orientation needed to kick off the wall. As a result, our approach achieved a much higher task return, with a value of 9.5 as indicated by the green curve in Figure 7(e).

Notably, as shown in Figure 7(f), $\lambda$ remained at 0 until reaching 0.5k steps and then gradually increased from 0 to 1 during the interval from 0.5k to 3k steps. In this experiment, $\lambda$ was capped at 1, which it eventually reached. Looking at the green training curves around the 3k step mark, the agent achieved a return of 5.0, indicating that it had reached the wall and needed to learn to kick off with its hind legs. If $\lambda$ had remained at 0, it would not have been able to achieve the necessary rotation, demonstrating that adjusting $\lambda$ has led to the acquisition of the specific orientation needed to kick off the wall.

## 5 CONCLUSION

In this work, we presented a novel framework that integrates unsupervised skill discovery with task-specific reinforcement learning to enable legged robots to learn highly agile locomotion behaviors with minimal manual intervention. By balancing exploration and task rewards through a bi-level optimization process, our method allows robots to discover diverse strategies and refine them to achieve complex tasks such as crawling, jumping, leaping, and performing agile maneuvers like wall-jumping.

We demonstrated that the incorporation of skill discovery methods not only facilitates the exploration of diverse behaviors but also enhances sample efficiency compared to traditional exploration-based techniques. We also showed that our method outperforms pure exploration-based baselines in various tasks, and the learned skills consistently converge to an optimal solution, ensuring the robustness and reproducibility of the learned behaviors. Furthermore, we pushed the boundary of agile locomotion learning with the successful implementation of the challenging wall-jump task, showcasing the potential of our method to handle even the most demanding dynamic behaviors. Future work could extend this approach to more diverse environments, exploring its potential in real-world robotic applications.

**Reproducibility Statement** We have made efforts to ensure the reproducibility of our work across various aspects.

- We provide detailed information about the observations, rewards coefficients, and hyperparameters used in our experiments in Appendix A.
- A comprehensive pseudo-code of our algorithm is available in Section 3.2.
- A thorough derivation of our method is presented in Section 3.2.
- Visualizations of the learned behaviors are presented in both Figure 5 and 7
- We present a video of our agents solving diverse tasks in supplementary material.
- We will also open-source the code if accepted.

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

# A IMPLEMENTATION DETAIL

## A.1 OBSERVATION SPACE

Table 2: A1 Robot Observations

| Name | Description | Dimension |
|------|-------------|-----------|
| Base position | x,y,z position of the robot's base | 3 |
| Base rotation | Yaw, Pitch, Roll of robot's base | 3 |
| Base velocity | velocity of robot's base in x,y,z direction | 3 |
| Base angvel | angular velocity of robot's base | 3 |
| Gravity projection | Vector indicates direction of the gravity | 3 |
| Velocity command | Velocity command given by users | 3 |
| DOF position | Current angle of each DOF | 12 |
| DOF velocity | Angular velocity of each DOF | 12 |
| Previous action | Action executed in previous step | 12 |
| Distance to obstacle | Distance to obstacle | 1 |
| Sidewall distance | Distance to side wall | 2 |
| Sampled Skill | Sampled skill for current episode | 2 |
| Sum | | 59 |

## A.2 TASK REWARD DETAIL

Table 3: Task rewards

| Name | Mathematical Expression | Coefficients value |
|------|------------------------|--------------------|
| Tracking angular velocity | $e^{-|w_{yaw}|}$ | 0.05 |
| Tracking linear velocity | $|v_x - v_x^{target}|$ | -1 |
| Alive | - | 2 |
| Torque squared | $\sum_{j \in joints} |\tau_j \dot{q}_j|^2$ | -1e-6 |
| Exceed dof pos limits | $\sum_{j \in joints} max(|dof_j| - dof_{lim}, 0)$ | -0.1 |
| Exceed torque limits | $\sum_{j \in joints} max(|\tau_j| - \tau_{lim}, 0)$ | -0.2 |

The first three terms about tracking commands specifies the goal of the task, while other three terms regularize unrealistic, infeasible motions.

## A.3 HYPERPARAMETERS

# B $\lambda$ CURVE FROM THREE PARKOUR LEARNING TASKS

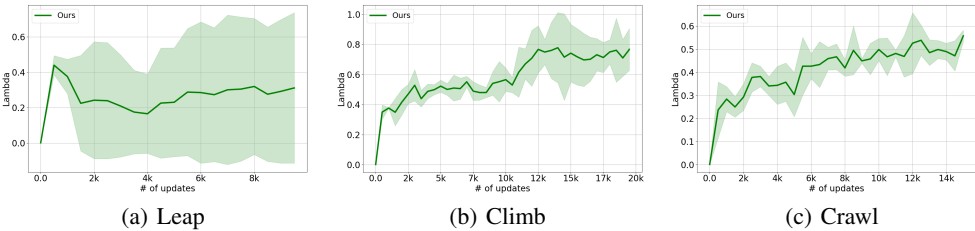

(a) Leap  (b) Climb  (c) Crawl

Figure 8: Different tasks yield different curve of $\lambda$.

Table 4: Hyperparameters of our method

| Name | Value |
|------|-------|
| Learning rate | 0.0005 |
| Optimizer | Adam(Kingma & Ba, 2014) |
| PPO clip threshold | 0.2 |
| PPO number of epochs | 5 |
| GAE $\lambda$ (Schulman et al., 2015) | 0.95 |
| Discount factor $\gamma$ | 0.99 |
| Horizon length | 24 |
| Entropy coefficient | 0.001 |
| Policy network $\pi$ | MLP with [512, 256, 128], |
| Activaion of $\pi$ | ELU(Clevert et al., 2015) |
| Value network $v$ | MLP with [512, 256, 128] |
| Activaion of $v$ | ELU(Clevert et al., 2015) |
| Representation function $\phi$ from Metra | MLP with [256, 256, 256] |
| Activaion of $\phi$ | ReLU |
| Initial Lagrange coefficient $\kappa$ from Metra | 30 |

For the climbing and crawling tasks, $\lambda$ gradually increases throughout training, reaching approximately 0.8 for climbing and 0.5 for crawling. In contrast, for the leaping task, $\lambda$ remains within the range of $[0.2, 0.4]$ without further increase. 3 different seeds were used.

## C  DETAILS OF THE GUIDELINE FOLLOWING REWARD

For the wall-jump task, we defined a special task reward, $r^{\text{task}}$, based on a guideline provided by a human. The guideline consists of a sequence of $n$ points:

$$g_{i=0,1,...,n-1} \in \mathbb{R}^3$$

Let the robot's base position in global 3D space be denoted as $\boldsymbol{x} \in \mathbb{R}^3$. At each time step, the robot has a target point $g_i$, starting with $g_0$. When the robot reaches the current target, it moves on to the next target, $g_{i+1}$. A target is considered reached when the distance between $\boldsymbol{x}$ and $g_i$ falls below a threshold $h \in \mathbb{R}$, i.e., $\|\boldsymbol{x} - g_i\|_2 < h$.

Then, the reward can be defined as follows:

$$r_t = e^{-\|\boldsymbol{x} - g_i\|_2}$$

This term has the desirable property of being bounded between 0 and 1. It approaches 0 when the robot is infinitely far from the current target and becomes 1 when the robot exactly reaches the target. This property contributes to stability during the learning process. We optimized this reward using reinforcement learning (RL) to train the agent to follow the given guideline.

