# OpenReview forum: "Interplay Between Task Learning and Skill Discovery for Agile Locomotion"
_ICLR.cc/2025/Conference — Submitted to ICLR 2025_

### Official Review · Reviewer_vxjR · 2024-10-29

**Soundness:** 3
**Presentation:** 3
**Contribution:** 3
**Rating:** 6
**Confidence:** 3

**Summary:**

This paper presents a novel RL framework for learning agile and diverse locomotion skills. This is achieved using the combination of maximising a task specific objective summed with a diversity one waited using a learnt parameter. The authors present a two stage training algorithm to optimise both the task and diversity objectives: The agent policy is updated using gradients from both objectives, whilst the learnt parameter is updated using only the task rewards. The algorithm in this paper is compared with a task-only, diversity-only (based on prior work METRA) and RND, which maximises reward exploration. Training with task and diversity objectives together outperform the base-lines and produces impressive results for three agile manoeuvres. These manoeuvres include leap, climb, crawl and a wall jump.

**Strengths:**

The formulation using a learnt weighting to balance the task and diversity objective is a strong contribution and is original. This is used to learn agile manoeuvres for quadruped locomotion in interesting environments and facilitates skill discovery. The quality of this work is generally good. The work is presented clearly and it is easy to follow. The work builds on METRA resulting in a contribution of merit to the field.

**Weaknesses:**

Despite the impressive results and soundness of the paper, there are a few weaknesses. These weaknesses are all readily addressable and addressing these would strengthen the paper. The most significant weaknesses are the comparisons to the baselines. The reason for using RND (Burda et al., 2018) as a baseline is not provided in the paper. RND should be included in the literature review and a few sentences explaining why this is a fair comparison should be provided Section 4.1. The METRA (div-only) baseline fails to learn a usable policy. It would be of benefit to the reader to explore why this fails and if the skill latent space learns anything useful. A potential weakness is that METRA and LSD permits the composition of skills albeit in a straightforward setting, is this possible with the task and diversity algorithm? The training curves in Fig. 7 e) and f) would be improved by showing the standard deviation across multiple repeats similarly to Fig. 4. Finally, a minor weakness is that Fig. 2 is of a slightly lower quality compared to the other figures.

**Questions:**

I have a few questions which mostly address some of the weaknesses of the paper.

Please could you point me to where you state how many repeats are used in Figure 4.

Please could you explain why there is such a broad range of obstacles successfully traversed in Figure 4.

As mentioned in weaknesses, why is RND used as a comparison baseline method? Why did the Div-only objective fail to solve the tasks? Did the latent representations of METRA fail to learn any reasonable skills?

Does the skill collapse mean that there is no smooth transition between skills?

Can you learn the skills needed for the wall jump individually and tie them together?

---

> ### Comment · Reviewer_vxjR · 2024-12-02
>
> I have not received any comments from the authors to answer my queries. As such I will not be changing my initial review scores.

---

### Official Review · Reviewer_dMLa · 2024-11-03

**Soundness:** 2
**Presentation:** 2
**Contribution:** 2
**Rating:** 5
**Confidence:** 4

**Summary:**

The paper proposes using a combination of skill discovery with goal-conditioned reinforcement learning, where the parameter balancing the two objectives is also optimized rather than held constant. The skill discovery is achieved by aligning latent space transitions with the sampled skills, subject to a temporal distance constraint. The approach allows learning agile behaviors for robot parkour with only a few simple task rewards, and the skill discovery module which encourages exploration. The method is verified on the quadruped robot A1 in simulation on a variety of tasks, including crawling, jumping, and wall-kicking. It outperforms the shown baselines and other exploration methods (RND) by a significant margin.

**Strengths:**

- Interesting application of skill discovery as a method of exploration in goal-conditioned policies, and applied to highly dynamic quadruped skills.

- The robot learns very diverse skills and the approach helps it converge quicker (or at all) to a solution, where using purely task rewards fails.|

- Simple task rewards result in impressive behaviors.

**Weaknesses:**

- Somewhat limited novelty - the approach uses an established skill discovery approach and balances its weight with task rewards, which prior work has already shown.

- No comparison with a few relevant prior works that also combine skill discovery with task policies.

- Lack of hardware validation, which makes it difficult to evaluate how useful this method can be in robotics.

- Some concerns with respect to the theoretical foundation of the method, as elaborated in the comments/questions.

**Questions:**

## Main questions/notes

1. Related work - the paper is missing some important references that propose very similar approaches, such as DOMiNO [2] and SMERL [1], which propose methods of combining task-rewards with learning diverse skills, using dual optimization with Lagrange multipliers, and DoMiNiC [3] which applies DOMiNO to quadruped tasks, in a very similar fashion to the proposed method.
I think a comparison with these methods would be highly beneficial to show the advantages of the proposed framework.


2. Hardware validation would make the contributions much stronger. One of the issues with skill discovery is that it can be potentially very jerky and jittery, and I think that showing whether such policies can safely translate to the hardware is important. For example, the wall-jump policy shown in the supplementary video looks very impressive, but I doubt it’d be easy to deploy on the hardware.

3. How smooth are the skills with respect to the behaviors? Are certain regions in the skill space more likely to cause failures than others, or is it entirely random?


4. Can METRA even work with goal-conditioned policies? The objective in METRA is formally defined as (phi(sT) - phi(s0))^T * z and is then reformulated as the telescopic sum that you have mentioned around line 220.  However, if you have a goal-conditioned objective (say reaching a point over the gap as in Fig. 5 a) or even a desired velocity), you’d essentially be constraining both the initial and final states (s0 and sT=s_goal). Since your initial state is fixed regardless of the skill, if you want to maximize the mutual information you’d need to vary your final state - and hence violate your goal objective. Other works get around that by maximizing the mutual information between the current state and skill (rather than the state difference as in METRA) or looking at the expected state distribution under skills
Can the proposed method then be seen as an approach for exploration rather than skill discovery? I think this should be made clearer in the paper. Since the skill discovery objective directly conflicts with the task objective, if I understand correctly, you would essentially be using skill discovery to explore diverse behaviors at the start of training, but still converge to more or less a single behavior at the end.


5. On that note, It would be interesting to see the METRA objective after training - is the alignment between latent transitions and skills still good?


6. Shouldn’t lambda start decreasing as you train your model further? After the agent has learned many skills, each with a different task return, wouldn’t the task reward be maximized if all skill-conditioned behaviors converge to the behavior corresponding to the skill with highest task return? From Fig 6/8 it seems that lambda is relatively stationary.

## Minor notes:

1. Section 3.1 mentions the action space as joint torques - is that really the case or is it a typo?


2. Fig. 6b) and Fig. 8 have “# of obstacles passed” as the y axis label - shouldn’t this be the value of lambda instead?


3. Does the METRA encoder use the same observations as the policy?

## Summary:
The paper proposes an interesting combination of skill discovery (through temporal distance maximization) and task rewards. The novelty seems to mostly come from the specific choice of skill discovery algorithm and the way the quality-diversity optimization is handled. The results overall seem promising, and I think some comparisons with competing methods like the ones mentioned above would strengthen the contributions. Furthermore, some hardware experiments would go a long way in showing how beneficial this approach is for robotics.

## Bibliography

[1]	S. Kumar, A. Kumar, S. Levine, and C. Finn, ‘One Solution is Not All You Need: Few-Shot Extrapolation via Structured MaxEnt RL’, in Advances in Neural Information Processing Systems, Curran Associates, Inc., 2020, pp. 8198–8210. doi: 10.48550/arXiv.2010.14484.

[2]	T. Zahavy et al., ‘Discovering Policies with DOMiNO: Diversity Optimization Maintaining Near Optimality’, presented at the The Eleventh International Conference on Learning Representations, Sep. 2022.

[3]	J. Cheng, M. Vlastelica, P. Kolev, C. Li, and G. Martius, ‘Learning Diverse Skills for Local Navigation under Multi-constraint Optimality’, in 2024 IEEE International Conference on Robotics and Automation (ICRA), Yokohama, Japan: IEEE, May 2024, pp. 5083–5089. doi: 10.1109/ICRA57147.2024.10611629.

---

> ### Author Response · Authors · 2024-11-24
>
> We thank the reviewer for their insightful review and for bringing up interesting discussion points. Please find our response below.
> ### Q1.1: Related Works
> Thank you for suggesting [1][2][3] as additional related works. [1] introduces a method for learning diverse behaviors to generalize across varying environments while accomplishing tasks. [2] employs a dual optimization approach with Lagrange multipliers to effectively balance the trade-off between quality and diversity. [3] builds on this by extending the approach to quadrupeds, enabling navigation around obstacles. These works are highly relevant to our work, and we will include them in the paper.
>
> ### Q1.2: Additional Baselines
> Thank you for suggesting additional baselines. We attempted to incorporate DoMiNiC [3] as an additional baseline but faced challenges with reward engineering, preventing successful integration into our environment. Instead, we have included DIAYN and Robot Parkour Learning in our evaluation. For DIAYN, we implemented a discrete version using PPO with a total of five discrete skills. This addition expands our evaluation to include exploration-based methods (RND), mutual information-based skill discovery methods (DIAYN), and distance maximization-based skill discovery methods (METRA).
> We also add Robot Parkour Learning as a specialized reinforcement learning baseline with a complex reward structure. We trained the parkour policy with soft dynamics for 70% of the training process, then fine-tuned it with hard dynamics for the remaining 30%, following the procedure outlined in the original paper. However, despite adhering to the reward terms and procedure described in the paper, we were unable to train a usable parkour policy for the Crawl task.
>
> For skill discovery methods, we sampled a new skill at the beginning of each episode and kept it fixed throughout that episode. For Ours (fixed z) we used a single fixed z value for all the episodes. The table below summarizes our findings, reporting the average number of obstacles passed (maximum of 3) over 100 episodes:
> | Method | Leap | Crawl | Climb |
> | :--- | :-----: | :-----: | :-----:|
> | Ours | 2.61 &plusmn; 1.01 | 0.98 &plusmn; 1.32 | 2.86 &plusmn; 0.62 |
> | Ours (fixed z) | **2.97 &plusmn; 0.30** | **1.89 &plusmn; 1.36** | **2.97 &plusmn; 0.30** |
> | DIAYN | 1.00 &plusmn; 0.00 | 0.00 &plusmn; 0.00 | 2.96 &plusmn; 0.32 |
> | RND | 2.83 &plusmn; 0.62 | 0.00 &plusmn; 0.00 | 0.00 &plusmn; 0.00 |
> | Div Only | 0.00 &plusmn; 0.00  | 0.04 &plusmn; 0.32 | 0.00 &plusmn; 0.00 |
> | Task Only | 0.00 &plusmn; 0.00 | 0.00 &plusmn; 0.00 | 0.00 &plusmn; 0.00 |
> | Parkour | 2.67 &plusmn; 0.90 | 0.00 &plusmn;0.00 | 2.76 &plusmn; 0.82 |
>
> From the table, we observe that our method performs on par with Robot Parkour Learning. This demonstrates that combining skill discovery with task rewards using our framework achieves approximately the same performance as a reinforcement learning policy with manually tuned reward functions but without the need for labor-intensive reward engineering. We also want to emphasize that the success rate of our method depends on the skills sampled at the start of each episode. These findings align with the observations reported in Table 1 of our main paper. We show that conditioning the policy on a fixed skill for all the eval episodes could enable our method to consistently perform well across all tasks. Notably we see a ~90% increase in the success rate for Crawl, and a near perfect success rate for Leap and Climb tasks outperforming even Robot Parkour Learning.
>
> Moreover, we find that DIAYN performs well on the Climb task but struggles with the Leap task and fails on the Crawl task. We attribute this to the nature of DIAYN's objective function, which maximizes mutual information (MI) using KL divergence. This approach often leads to less distinctive behaviors, as KL divergence is fully maximized when two distributions have no overlap. Beyond this point, there is no additional incentive to further distinguish between distributions. As a result, DIAYN suffers from poor exploration compared to distance-maximization-based skill discovery methods, which are better suited for generating diverse and effective skills.
>
> [1] S. Kumar, A. Kumar, S. Levine, and C. Finn, ‘One Solution is Not All You Need: Few-Shot Extrapolation via Structured MaxEnt RL’, in Advances in Neural Information Processing Systems, Curran Associates, Inc., 2020, pp. 8198–8210. doi: 10.48550/arXiv.2010.14484.
>
> [2] T. Zahavy et al., ‘Discovering Policies with DOMiNO: Diversity Optimization Maintaining Near Optimality’, presented at the The Eleventh International Conference on Learning Representations, Sep. 2022.
>
> [3] J. Cheng, M. Vlastelica, P. Kolev, C. Li, and G. Martius, ‘Learning Diverse Skills for Local Navigation under Multi-constraint Optimality’, in 2024 IEEE International Conference on Robotics and Automation (ICRA), Yokohama, Japan: IEEE, May 2024, pp. 5083–5089. doi: 10.1109/ICRA57147.2024.10611629.

---

> ### Author Response · Authors · 2024-11-24
>
> ### Q2: Hardware Validation
> We appreciate the reviewer’s suggestion to include hardware results and their concerns about unnatural motion. However, due to time constraints and the labor-intensive nature of hardware experiments, we are unable to provide hardware results for our method at this stage. While our approach generally produces natural-looking motions, we acknowledge that some tasks may not directly translate to real-world applications. The primary objective of this work is to demonstrate an effective method for combining skill discovery techniques with task rewards, eliminating the need for additional approaches like reward engineering to achieve agile motions. Investigating the applicability of our method to real-world robotic tasks remains an exciting avenue for future research.
>
>
> ### Q3: Smoothness of skills and distribution of successful skills in the skill space.
>
> Thank you for this insightful question, which prompted us to perform additional analysis. To evaluate the smoothness of the skills, we conducted a simple experiment on the Crawl task. Specifically, we manually identified two skill latent values:  $z_{fail}$, corresponding to behaviors that fail to complete the task, and $z_{success}$, corresponding to behaviors that successfully complete it.
> For each of these values, we sampled 10 skills from the ranges $z_{fail}$ &plusmn;0.05 and $z_{success}$ &plusmn;0.05 , with intervals of 0.01, as our skill latent space is one-dimensional. The table below shows the average number of obstacles passed for these 10 sampled skills, providing insight into the smoothness of the skill transitions.
> |  | $z_{fail}$| $z_{success}$ |
> | :--- | :-----: | :-----: |
> |Crawl | 0.00 &plusmn; 0.00 | 1.90 &plusmn; 1.45 |
>
>
> We observe that all the skills sampled near $z_{fail}$ result in failure, while those in the vicinity of $z_{success}$ consistently succeed, as shown in the results above. This indicates that the skills are smooth with respect to their corresponding behaviors.
>
>
> ### Q4: Can METRA work with goal conditioned policices?
>
> Yes, we believe METRA can work effectively with goal-conditioned policies. We understand your concern, but METRA and goal-conditioned policies operate on *different sub-dimensions* of the state space. For instance, in the crawl task, we use the linear velocity of the robot's base as the goal, while simultaneously using the robot's base height as input to METRA. These two sub-dimensions do not overlap. In other words, the robot can follow a specific linear velocity while exploring diverse heights of its base. By trying out different base heights, the robot discovers a solution to the task—crawling.
> Furthermore, even when sub-dimensions overlap, METRA and goal-conditioned policies can still complement each other. For example, in the leap task, consistently following the goal speed of 0.7 m/s would prevent the robot from successfully jumping over the gap. To make the jump, the robot must deviate from the goal speed momentarily to achieve the necessary dynamics, allowing it to continue following the desired speed for the rest of the episode. Without this deviation, the robot fails to clear the gap and crashes. In this scenario, the task reward guides the policy to adhere to the goal velocity as much as possible, while METRA enables the robot to explore different velocities when required.
> Lastly, we confirm that we used a skill discovery framework as our exploration method. We have clarified this in lines 315–319.
>
> ### Q6: Shouldn’t lambda start decreasing as the model trains further?
>
> First, intuitively, the value of λ should decrease when less exploration is needed. This assumes that the diversity reward should work as an exploration module. If the diversity reward does not serve as an exploration module, there would be no incentive to reduce the value of λ. This is indeed the case in our approach. As training progresses and most skills converge into a narrow solution space, exploring different skills tends to produce similar behaviors and task rewards. We refer to this phenomenon as “positive skill collapse,” which we discuss in detail in Section 4.3.
>
>
> In summary, since different skills no longer produce diverse behaviors at the later stage of training, there is no incentive to decrease the λ value.
>
> ### M1: Section 3.1
> This is indeed a typo, the action space is pertubations in the joint angles form the default position. We will fix this is in the paper.
>
>
> ### M2: Figure 6b) and Fig 8.
> Thank you for pointing out the mislabelling in the plots. We have fixed them in the paper.
>
>
> ### M3: Does the METRA encoder use the same observations as the policy?
> No, the METRA encoder only gets certain states as inputs. These vary for different tasks. For Climb and Crawl tasks we use robot height and in Leap we use velocity in forward direction as the input to METRA encoder. We also mention this in Section 4.1, line 304.

---

> > ### Comment · Reviewer_dMLa · 2024-11-30
> >
> > Thank you for the efforts in addressing the concerns in the original review. I think the additional comparison with Robot Parkour Learning makes the contributions stronger. Albeit the increase in success rate is minor, it is advantageous than the proposed method has much fewer reward terms. However, I'm still not fully convinced on the utility of the method without any hardware validation. For example, some of the reward terms in baselines (like Robot Parkour Learning) might trade-off simulation for hardware performance, and so it is hard to judge the slightly better success rate of the proposed method in sim only.

---

### Official Review · Reviewer_45ih · 2024-11-04

**Soundness:** 2
**Presentation:** 3
**Contribution:** 2
**Rating:** 3
**Confidence:** 4

**Summary:**

This paper combines reinforcement learning and unsupervised skill discovery to learn agile locomotion. A bi-level optimization is proposed to train a parameter to balance the task reward and the diversity reward. The proposed approach can achieve impressive locomotion task, such as a simulated wall-jump.

**Strengths:**

- This paper clearly presents the contributions and results.
- The achieved wall-jump in simulation looks impressive.

**Weaknesses:**

- The novelty of this paper is limited.
- More baselines should be included, e.g., a pure RL policy with more complex task reward such as those parkour policies.
- Some of the learned motions look a bit unrealistic. Real-world experiments would strengthen the contribution of the proposed approach.
- More task-related plots should be included, e.g., body orientation plots, joint torque plots etc.
- For videos, it would be better to include failure cases, baselines. The video shows diverse solutions to one task, but should also show something like jumping over or crawling under the same hanging obstacle as in Fig. 2.

**Questions:**

- Detailed formulation of task reward is missing. Mathematical formulations and the desired values for calculating the task reward for each task need to be provided.
- Why is the standard deviation of the training curve of the proposed approach so large, e.g., Fig. 4a & 4b?
- The standard deviation of the lambda curve for leaping task in Fig. 8 is very large. Why the authors conclude it is always within the range of 0.2-0.4? Besides, y-axis name is not correct for some lambda curves throughout the paper, e.g., Fig. 6b.
- When comparing the proposed approach with fixed lambda values, the constant lambda values should include the converged values as well.

---

> ### Author Response · Authors · 2024-11-24
>
> Thank you for your thoughtful review. Below, we provide detailed answers to your comments.
>
>
> ### W2: Additional Baselines
> Thank you for suggesting additional baselines. In response, we have incorporated both DIAYN and Robot Parkour Learning into our evaluation. For DIAYN, we implemented a discrete version using PPO with a total of five discrete skills. This addition expands our evaluation to include exploration-based methods (RND), mutual information-based skill discovery methods (DIAYN), and distance maximization-based skill discovery methods (METRA).
> We also added Robot Parkour Learning as a specialized reinforcement learning baseline with a complex reward structure. We trained the parkour policy with soft dynamics for 70% of the training process, then fine-tuned it with hard dynamics for the remaining 30%, following the procedure outlined in the original paper. However, we were not able to train a usable parkour policy for Crawl even after following the reward terms and process as mentioned in the paper.
>
>
> For skill discovery methods, we sampled a new skill at the beginning of each episode and kept it fixed throughout that episode. For Ours (fixed z) we used a single fixed z value for all the episodes. The table below summarizes our findings, reporting the average number of obstacles passed (maximum of 3) over 100 episodes:
> | Method | Leap | Crawl | Climb |
> | :--- | :-----: | :-----: | :-----:|
> | Ours | 2.61 &plusmn; 1.01 | 0.98 &plusmn; 1.32 | 2.86 &plusmn; 0.62 |
> | Ours (fixed z) | **2.97 &plusmn; 0.30** | **1.89 &plusmn; 1.36** | **2.97 &plusmn; 0.30** |
> | DIAYN | 1.00 &plusmn; 0.00 | 0.00 &plusmn; 0.00 | 2.96 &plusmn; 0.32 |
> | RND | 2.83 &plusmn; 0.62 | 0.00 &plusmn; 0.00 | 0.00 &plusmn; 0.00 |
> | Div Only | 0.00 &plusmn; 0.00  | 0.04 &plusmn; 0.32 | 0.00 &plusmn; 0.00 |
> | Task Only | 0.00 &plusmn; 0.00 | 0.00 &plusmn; 0.00 | 0.00 &plusmn; 0.00 |
> | Parkour | 2.67 &plusmn; 0.90 | 0.00 &plusmn;0.00 | 2.76 &plusmn; 0.82 |
>
> From the table, we observe that our method performs on par with Robot Parkour Learning. This demonstrates that combining skill discovery with task rewards using our framework achieves approximately the same performance as a reinforcement learning policy with manually tuned reward functions, but without the need for labor-intensive reward engineering. We also want to emphasize that the success rate of our method depends on the skills sampled at the start of each episode. These findings align with the observations reported in Table 1 of our main paper. We show that conditioning the policy on a fixed skill for all the eval episodes could enable our method to consistently perform well across all tasks. Notably we see a ~90% increase in the success rate for Crawl, and a near perfect success rate for Leap and Climb tasks outperforming even Robot Parkour Learning.
>
>
> Through our evaluation, we find that DIAYN performs well on the Climb task but struggles with the Leap task and completely fails on the Crawl task. We attribute this to the nature of DIAYN's objective function, which maximizes mutual information (MI) using KL divergence. This approach often leads to less distinctive behaviors, as KL divergence is fully maximized when two distributions have no overlap. Beyond this point, there is no additional incentive to further distinguish between distributions. As a result, DIAYN suffers from poor exploration compared to distance-maximization-based skill discovery methods, which are better suited for generating diverse and effective skills.
>
>
> ### W3: Hardware Validation and Unnatural Motion
>
>
> We appreciate the reviewer’s suggestion to include hardware results and their concerns about unnatural motion. However, due to time constraints and the labor-intensive nature of hardware experiments, we are unable to provide hardware results for our method at this stage. While our approach generally produces natural-looking motions, we acknowledge that some tasks may not directly translate to real-world applications. The primary objective of this work is to demonstrate an effective method for combining skill discovery techniques with task rewards, eliminating the need for additional approaches like reward engineering to achieve agile motions. Investigating the applicability of our method to real-world robotic tasks remains an exciting avenue for future research.
>
>
> ### W4 & 5: Task-related plots and Failure cases in Videos
> Thank you for your suggestion. We kindly request the reviewer to clarify the specific type of plots they would like us to include. Additionally, we will incorporate more video experiments in line with the reviewer's recommendation.

---

> > ### Comment · Reviewer_45ih · 2024-11-26
> >
> > Thank you for comparing with additional baselines and the efforts for revising the paper. However, I'm not convinced of the novelty of this paper in its current form due to the following reasons:
> >
> > W1: The authors did not address this weakness regarding limited novelty. I do not see significant improvement of novelty after revision either.
> >
> > W3: I do not agree with the authors' statement "Our approach generally produces natural-looking motions...". Some jumping motions on staircases and the walking before wall-jump do not look natural. Since the paper has limited technical novelty, I assume the application on robots would be the major novelty. However, other skill discovery paper have already shown natural quadruped locomotion on hardware. Without hardware experiments in this paper, I'm not convinced of the contributions.
> >
> > W4 & 5: The authors should at least show body orientation plot and joint torque plot for the wall-jump task. For certain robot poses, the joint torque might exceed the limit already, which needs to be reported. Besides, I do not see any additional experiments in the video.

---

> ### Author Response · Authors · 2024-11-24
>
> ### Q1: Detailed formulation of rewards
> We have added the mathematical formulation of the rewards in Appendix A.2.
>
>
> ### Q2: Standard deviation being too high
> The high standard deviation in our results can be attributed to the following factors:
>
>
> i) Discrete Metric for Evaluation: We use the number of obstacles passed as a discrete performance metric, which inherently introduces variability. Since this metric represents distinct thresholds rather than a continuous range, small differences in performance can result in significant deviations in the reported averages.
>
>
> ii) Variance in Convergence Across Seeds: Our method exhibits variability in the number of training epochs required to converge across different random seeds. This inconsistency in convergence times contributes to fluctuations in the performance metrics, leading to a higher standard deviation.
>
>
> ### Q3: Lambda Curve Fig. 8
>
>
> We would like to clarify that when we mentioned the lambda values staying within the range of 0.2–0.4, we were specifically referring to the **mean** of the lambda values. The primary point we intended to emphasize was the relatively constant behavior of lambda during the Leap task while it is constantly increasing for the Crawl and Climb tasks. This variation highlights how lambda adapts differently depending on the task dynamics and requirements.
>
>
> To avoid any ambiguity, we will revise the text in the paper to make this explanation clearer and more precise.
>
>
> ### Q4: Comparison against fixed lambda value
>
>
> We would first like to clarify that we do not consider a single optimal λ value to exist for the entire training process. Instead, the purpose of adapting λ during training is not to find one "correct" value, but to dynamically adjust it based on the current training phase.
>
>
> For example, at the beginning of training, λ should remain small so that the agent focuses solely on the task reward. This is beneficial because the agent can develop a straightforward forward-moving policy without requiring additional exploration. However, when the agent encounters an obstacle, λ should increase to encourage exploration and allow the agent to discover a solution. If λ were large from the beginning of the episode, the agent might not focus effectively on maximizing the task reward. Similarly, as the task is being solved, the optimal value of λ may change depending on the requirements of that phase.
>
>
> In summary, the optimal λ value varies across different phases of training, and our method provides a way to automatically adjust λ to maximize task rewards at each stage. Therefore, we do not aim to find a single λ value that performs best across all stages of training. This is why we did not include the final λ value from the training run in our results.
>
>
> That said, if you are curious, we conducted additional experiments using the final λ value from the "crawl" task (0.58) and reran the experiments with a fixed λ value of 0.58. We have provided the plot in the supplementary material. Although our method was able to solve the task, using a fixed λ value of 0.58 introduced challenges in solving the task effectively. If needed, we are willing to run further experiments with corresponding fixed λ values for other tasks.

---

### Official Review · Reviewer_RNtf · 2024-11-04

**Soundness:** 2
**Presentation:** 2
**Contribution:** 2
**Rating:** 3
**Confidence:** 4

**Summary:**

This study introduces a method for teaching robots agile locomotion skills using unsupervised reinforcement learning (RL). The proposed RL approach combines a task-specific reward with an unsupervised skill discovery reward, scaled dynamically. This combination allows the robot to acquire skills such as jumping, running, and flipping without specially designed rewards or expert demonstrations. The approach was evaluated in simulation, showing reasonable performance improvements over existing baselines.

**Strengths:**

1. The method is intuitive and appears straightforward to implement, with detailed information provided for reproducing results in simulation.
2. The approach successfully learned complex locomotion skills, including wall-jumping, within a simulated environment.
3. The ablation study shows automatically tune the weight for unsupervised rl objective helps on improving the sample efficiency.

**Weaknesses:**

1. The method is tested solely in simulation without detailing specific simulation physics parameters. Given recent advancements in robot learning for quadrupedal robots, a real-world evaluation would help strengthen the impact and credibility of the results.

2. The motions generated by the learned model appear unnatural, which may hinder transferability to real-world applications.

3. Experimental setup and evaluation raise several concerns:

* Evaluations were conducted in three relatively simple environments (Leap, Climb, Crawl), none of which are as challenging as the demonstrated wall-jump skill.
* The study compares only with limited baselines (Task-Only, Div-Only, RND), whereas it would be informative to include diversity-oriented baselines such as DIAYN, as seen in previous work like METRA (Park, 2023).
* The task scope closely resembles that of Robot Parkour Learning (Zhuang 2023), so comparing it with methods that use manually tuned reward functions would help clarify the specific contributions of this work.
* The wall-jump task appears to have been run with a single seed and without robust baseline comparisons across other tasks, so further clarification is needed here.

4. Some related literature is missing:

Agile Locomotion:

"Lifelike agility and play on quadrupedal robots using reinforcement learning and generative pre-trained models"

"Generalized Animal Imitator: Agile Locomotion with Versatile Motion Prior"

Unsupervised RL for Learning Locomotion Skills:

"ASE: Large-scale reusable adversarial skill embeddings for physically simulated characters"

Using Auxiliary Rewards for Enhancing Robot Learning:

"Adversarial Motion Priors Make Good Substitutes for Complex Reward Functions"

**Questions:**

Since the proposed method is applying existing unsupervised RL work to the robotics tasks, more experiments should be conducted to make it a promising submission. Please refer to the weakness mentioned above

---

> ### Author Response · Authors · 2024-11-24
>
> We thank the reviewer for the constructive feedback and suggestions to improve our work. Please find our answers to the questions below.
> ### W1, W2: Hardware Validation, Unnatural Motion and physics paramters
>
> We appreciate the reviewer’s suggestion to include hardware results and their concerns about unnatural motion. However, due to time constraints and the labor-intensive nature of hardware experiments, we are unable to provide hardware results for our method at this stage. While our approach generally produces natural-looking motions, we acknowledge that some tasks may not directly translate to real-world applications. The primary objective of this work is to demonstrate an effective method for combining skill discovery techniques with task rewards, eliminating the need for additional approaches like reward engineering to achieve agile motions. Investigating the applicability of our method to real-world robotic tasks remains an exciting avenue for future research.
>
>
> We will add the details about the physics parameters in the appendix of the paper
>
>
> ### W3.1: Evaluation environments
> As the reviewer mentioned, we evaluated our methods on Leap, Crawl, and Climb tasks as well as on a challenging wall-jump task. We followed Robot Parkour Learning's task environment for evaluation. We will be happy to evaluate on more difficult tasks on the reviewer's suggestion.

---

> ### Author Response · Authors · 2024-11-24
>
> ### W3.2: Additional Baselines
> Thank you for suggesting additional baselines. In response, we have incorporated both DIAYN and Robot Parkour Learning into our evaluation. For DIAYN, we implemented a discrete version using PPO with a total of five discrete skills. This addition expands our evaluation to include exploration-based methods (RND), mutual information-based skill discovery methods (DIAYN), and distance maximization-based skill discovery methods (METRA).
> We also added Robot Parkour Learning as a specialized reinforcement learning baseline with a complex reward structure. We trained the parkour policy with soft dynamics for 70% of the training process, then fine-tuned it with hard dynamics for the remaining 30%, following the procedure outlined in the original paper. However, we were not able to train a usable parkour policy for Crawl even after following the reward terms and process as mentioned in the paper.
>
>
> For skill discovery methods, we sampled a new skill at the beginning of each episode and kept it fixed throughout that episode. For Ours (fixed z) we used a single fixed z value for all the episodes. The table below summarizes our findings, reporting the average number of obstacles passed (maximum of 3) over 100 episodes:
> | Method | Leap | Crawl | Climb |
> | :--- | :-----: | :-----: | :-----:|
> | Ours | 2.61 &plusmn; 1.01 | 0.98 &plusmn; 1.32 | 2.86 &plusmn; 0.62 |
> | Ours (fixed z) | **2.97 &plusmn; 0.30** | **1.89 &plusmn; 1.36** | **2.97 &plusmn; 0.30** |
> | DIAYN | 1.00 &plusmn; 0.00 | 0.00 &plusmn; 0.00 | 2.96 &plusmn; 0.32 |
> | RND | 2.83 &plusmn; 0.62 | 0.00 &plusmn; 0.00 | 0.00 &plusmn; 0.00 |
> | Div Only | 0.00 &plusmn; 0.00  | 0.04 &plusmn; 0.32 | 0.00 &plusmn; 0.00 |
> | Task Only | 0.00 &plusmn; 0.00 | 0.00 &plusmn; 0.00 | 0.00 &plusmn; 0.00 |
> | Parkour | 2.67 &plusmn; 0.90 | 0.00 &plusmn;0.00 | 2.76 &plusmn; 0.82 |
>
>
> From the table, we observe that our method performs on par with Robot Parkour Learning. This demonstrates that combining skill discovery with task rewards using our framework achieves approximately the same performance as a reinforcement learning policy with manually tuned reward functions, but without the need for labor-intensive reward engineering. We also want to emphasize that the success rate of our method depends on the skills sampled at the start of each episode. These findings align with the observations reported in Table 1 of our main paper. We show that conditioning the policy on a fixed skill for all the eval episodes could enable our method to consistently perform well across all tasks. Notably we see a ~90% increase in the success rate for Crawl, and a near perfect success rate for Leap and Climb tasks outperforming even Robot Parkour Learning.
>
>
> Through our evaluation, we find that DIAYN performs well on the Climb task but struggles with the Leap task and completely fails on the Crawl task. We attribute this to the nature of DIAYN's objective function, which maximizes mutual information (MI) using KL divergence. This approach often leads to less distinctive behaviors, as KL divergence is fully maximized when two distributions have no overlap. Beyond this point, there is no additional incentive to further distinguish between distributions. As a result, DIAYN suffers from poor exploration compared to distance-maximization-based skill discovery methods, which are better suited for generating diverse and effective skills.
>
>
>
>
> Due to time and computational constraints, we are unable to conduct additional experiments with more seeds and baseline comparisons for the Wall-Jump task at this time. However, we will ensure that these additions are included in the final version of the paper.
>
>
> ### W4: Related Works
>
>
> We appreciate the reviewer’s suggestions and agree that [1][2] are highly relevant works leveraging motion priors to achieve agile locomotion. Similarly, [3][4] are valuable contributions that aim to combine unsupervised reinforcement learning and imitation learning to address the challenge of complex reward engineering. We will update the related works section of our paper to include these references.
>
>
> [1] Han, Lei, et al. "Lifelike agility and play in quadrupedal robots using reinforcement learning and generative pre-trained models." Nature Machine Intelligence 6.7 (2024): 787-798.
>
>
> [2] Yang, Ruihan, et al. "Generalized animal imitator: Agile locomotion with versatile motion prior." arXiv preprint arXiv:2310.01408 (2023).
>
>
> [3] Peng, Xue Bin, et al. "Ase: Large-scale reusable adversarial skill embeddings for physically simulated characters." ACM Transactions On Graphics (TOG) 41.4 (2022): 1-17.
>
>
> [4] Escontrela, Alejandro, et al. "Adversarial motion priors make good substitutes for complex reward functions." 2022 IEEE/RSJ International Conference on Intelligent Robots and Systems (IROS). IEEE, 2022.

---

> ### Comment · Reviewer_RNtf · 2024-11-26
>
> Thank you for the efforts in providing additional comparisons against baselines. However, I am keeping my original evaluation of the work, primarily due to the absence of real robot evaluation. Given the recent developments in locomotion research, it has become standard for most works, including agile locomotion methods, to demonstrate their approach on real hardware. Considering the limited novelty of the proposed method, real robot evaluation is essential to justify publication.
>
> Additionally, I have the following minor concerns: Unnatural Motion: The generated motions appear unnatural in certain scenarios, which raises questions about the applicability of the method to real-world tasks.
> Lack of Multi-Seed Results: For the most challenging task, the wall-jump, results have not been evaluated across multiple seeds. This makes it difficult to assess the robustness and consistency of the proposed method.

---

### Meta-Review · Area_Chair_r8s7 · 2024-12-20

**Metareview:**

(a) The paper introduces a framework combining reinforcement learning with unsupervised skill discovery to enable legged robots to learn agile locomotion skills. The core claim is that by using a learnable parameter to balance task-specific rewards with a diversity reward, robots can learn complex motions without extensive reward engineering, expert demonstrations, or curriculum learning. The method can train robots to perform highly agile motions, such as crawling, jumping, leaping, and a wall-jump, achieving performance on par with or better than manually tuned reward systems, without intensive reward engineering.

(b) Strengths:
- The method successfully trains robots to perform complex, agile motions such as wall-jumping with good performance in sim.
- The paper is generally well-written and easy to follow, with detailed information for reproducing results in simulation.
- The methodology description is comprehensive.

(c) Weaknesses:
- Limited Novelty: Reviewers noted that the method combines existing techniques, and thus the overall novelty is limited.
- No real-world experiment. Some motions appear unnatural, which may limit the transferability to the real world.

(d) The decision is to reject the paper. For locomotion research paper, the weaknesses in the lack of real-world experiment and methodological novelty are significant.

**Additional Comments On Reviewer Discussion:**

Novelty: Several reviewers noted that the paper had limited novelty, as it combines existing skill discovery and reinforcement learning techniques. The authors acknowledged this but emphasized that the primary contribution is demonstrating an effective method for combining skill discovery with task rewards to achieve agile locomotion without extensive reward engineering. They also highlighted that their approach allows for the learning of agile motions, including the wall-jump, without manually designed rewards. However, the reviewers are not fully convinced.

Lack of Real-World Experiments: Reviewers expressed concerns about the absence of real-world experiments, arguing that simulation results alone are insufficient to prove the practical applicability of the method. The authors explained that due to time and resource limitations, they could not conduct real-world experiments at this time.

Unnatural Motions: Reviewers pointed out that some of the generated motions appeared unnatural, raising concerns about the transferability. The authors acknowledged that some motions might not directly translate to real-world applications, but reiterated their focus on combining skill discovery with task rewards.

Limited Baselines: Reviewers criticized the limited baseline comparisons, suggesting comparing to DIAYN and manually tuned reward functions as in Robot Parkour Learning, for example. The authors added DIAYN and Robot Parkour Learning as baselines, which satisfied the reviewers. Although the authors incorporated additional baselines, some reviewers felt the comparison was limited and did not include all relevant methods.

The reviewers mostly maintained their initial scores, leaning to rejection, as the above weaknesses are not (fully) resolved. I agree with the reviewers.

---

### Decision · Program_Chairs · 2025-01-22

Reject